# Negative magnetoresistance without well-defined chirality in the Weyl semimetal TaP

Frank Arnold[1,*], Chandra Shekhar[1,*], Shu-Chun Wu[1], Yan Sun[1], Ricardo Donizeth dos Reis[1], Nitesh Kumar[1], Marcel Naumann[1], Mukkattu O. Ajeesh[1], Marcus Schmidt[1], Adolfo G. Grushin[2], Jens H. Bardarson[2], Michael Baenitz[1], Dmitry Sokolov[1], Horst Borrmann[1], Michael Nicklas[1], Claudia Felser[1], Elena Hassinger[1] & Binghai Yan[1,2]

Weyl semimetals (WSMs) are topological quantum states wherein the electronic bands disperse linearly around pairs of nodes with fixed chirality, the Weyl points. In WSMs, nonorthogonal electric and magnetic fields induce an exotic phenomenon known as the chiral anomaly, resulting in an unconventional negative longitudinal magnetoresistance, the chiral-magnetic effect. However, it remains an open question to which extent this effect survives when chirality is not well-defined. Here, we establish the detailed Fermi-surface topology of the recently identified WSM TaP via combined angle-resolved quantum-oscillation spectra and band-structure calculations. The Fermi surface forms banana-shaped electron and hole pockets surrounding pairs of Weyl points. Although this means that chirality is ill-defined in TaP, we observe a large negative longitudinal magnetoresistance. We show that the magnetoresistance can be affected by a magnetic field-induced inhomogeneous current distribution inside the sample.

[1] Max Planck Institute for Chemical Physics of Solids, Dresden 01187, Germany. [2] Max Planck Institute for the Physics of Complex Systems, Dresden 01187, Germany. * These authors contributed equally to this work. Correspondence and requests for materials should be addressed to E.H. (email: elena.hassinger@cpfs.mpg.de) or to B.Y. (email: yan@cpfs.mpg.de).

In a semimetal the conduction and valence bands touch at isolated points in the three-dimensional (3D) momentum ($k$) space at which the bands disperse linearly. Depending on whether the bands are nondegenerate or doubly degenerate, such a 3D semimetal is called a Weyl semimetal (WSM)[1–9] or a Dirac semimetal[10–12], respectively. Correspondingly, the band touching point is referred to as a Weyl or a Dirac point. The Dirac point can split into one or two pairs of Weyl points by breaking either time-reversal symmetry or crystal inversion symmetry. At energies close to the Weyl points, electrons behave effectively as Weyl fermions, a fundamental kind of massless fermions that has never been observed as an elementary particle[13]. In condensed-matter physics, each Weyl point acts as a singularity of the Berry curvature in the Brillouin zone (BZ), equivalent to magnetic monopoles in $k$ space. Thus Weyl points always occur in pairs with opposite chirality or handedness[14]. In the presence of nonorthogonal magnetic ($\mathbf{B}$) and electric ($\mathbf{E}$) fields (that is, $\mathbf{E} \cdot \mathbf{B}$ is nonzero), the particle number for a given chirality is not conserved quantum mechanically, inducing a phenomenon known as the Adler–Bell–Jackiw anomaly or chiral anomaly in high-energy physics[13,15,16]. In WSMs, the chiral anomaly is predicted to lead to a negative magnetoresistance (MR) due to the suppressed backscattering of electrons of opposite chirality[17,18]. Theoretically, the chiral anomaly only appears, if chirality is well-defined, that is, the Fermi energy is close enough to the Weyl nodes that pairs of separate Fermi surface pockets with opposite chirality exist[17]. Observing the chiral-anomaly induced negative MR requires the applied magnetic and electric field to be as parallel as possible. Otherwise the negative MR will easily be overwhelmed by the positive contribution arising due to the Lorentz force. In addition to the negative MR, the chiral anomaly is also predicted to induce an anomalous Hall effect[2,19–21], nonlocal transport properties[22,23] and sharp discontinuities in angle-resolved photo-emission spectroscopy (ARPES) signals[24] in WSMs.

The discovery of various WSM materials has stimulated experimental efforts to confirm the chiral anomaly in condensed-matter physics. Recently, a negative MR has been reported in two types of WSMs: WSMs induced by time-reversal symmetry breaking, that is, Dirac semi metals in an applied magnetic field, for example $Bi_{1-x}Sb_x$ ($x \approx 3\%$)[25], $ZrTe_5$ (ref. 26) and $Na_3Bi$[27], and the non-inversion-symmetric WSMs TaAs[28,29], NbP[30] and NbAs[31]. However, a clear verification of whether the Fermi surface topology supports the chiral anomaly or not, is still lacking in most of the above systems.

In the non-centrosymmetric WSMs of the TaAs family two types of Weyl nodes exist at different positions in reciprocal space[5,6] and energies. Therefore, the Weyl electrons generally coexist with topologigcally trivial normal electrons. In principle, small changes of the Fermi energy ($E_F$), as induced by doping or defects, can change the Fermi-surface topology significantly due to the low intrinsic charge carrier density in semimetals. Therefore, a precise knowledge of $E_F$ and the resulting Fermi-surface topology is required when linking the negative MR to the chiral-magnetic effect. Extensive ARPES studies have shown the existence of Fermi-arc surface states and linear band crossings in the bulk band structure of all four materials from the TaAs family[7–9,32,33]. However, because of an insufficient energy resolution ($>15$ meV (ref. 32)), ARPES is not able to make any claims about the presence or absence of quasiparticles with well-defined chirality at the Fermi level. In contrast, quantum-oscillation measurements have the advantage of a millielectronvolt resolution of the Fermi-energy level.

In this work, we reconstruct the 3D Fermi surface of TaP by combining sensitive Shubnikov-de Haas (SdH) and de Haas-van Alphen (dHvA) oscillations with *ab initio* band-structure

calculations reaching a good agreement between theory and experiment. We reveal that $E_F$ is such that the electron and hole Fermi-surface pockets contain pairs of Weyl nodes and the total Berry flux through the Fermi surface $\Phi_B = 0$. Although this means that chirality is not well-defined, a large negative MR is observed. We discuss possible explanations for this result also considering the magnetic field dependence of the current distribution in our samples[34].

## Results

**Single-crystal synthesis and characterization.** We synthesized high-quality single crystals of TaP by using chemical vapour transport reactions and verified TaP as a non-centrosymmetric compound in a tetragonal lattice (space group I4$_1$md, No. 109). The temperature-dependent resistivity exhibits typical semimetallic behaviour. For more details, see Methods and Supplementary Figs 1–4.

**Quantum oscillations.** The Fermi surface topology of TaP was investigated by means of quantum oscillations. Typically, for a semimetal with light carriers and high mobility, such as bismuth[35], prominent oscillations appear in all properties sensitive to the density of states at the Fermi energy. Here, we measured SdH oscillations in transport (Fig. 1a) and dHvA oscillations in magnetic torque (Fig. 1b) and magnetization (Fig. 2a) for different magnetic field orientations. These oscillations are periodic in $1/B$. Their frequency ($F$) is proportional to the corresponding extremal Fermi surface cross-section ($A$) that is perpendicular to $\mathbf{B}$ following the Onsager relation $F = (\Phi_0/2\pi^2)A$, where $\Phi_0 = h/2e$ is the magnetic flux quantum and $h$ is the Planck constant. Figure 1a shows the resistivity as a function of the magnetic field for different field orientations. When the electric current and magnetic field are perpendicular ($\mathbf{I}\|\mathbf{a}, \mathbf{B}\|\mathbf{c}$), the magnetoresistance is very high as typical for other WSM (for example, ref. 36) and normal semimetals[37,38]. This implies a very high mobility of the charge carriers. Figure 1b depicts the magnetic torque oscillations for the same field orientations. Figure 2a represents the magnetic dHvA oscillations in the magnetization as a function of the inverse magnetic field and their corresponding Fourier transform (Fig. 2b) for $\mathbf{B}\|\mathbf{c}$. The observable fundamental frequencies are $F_\alpha = 15$ T, $F_\beta = 18$ T, $F_\gamma = 25$ T, and $F_\delta = 45$ T. These frequencies are consistent between all three measurement techniques and different sample batches within the error bars (Supplementary Table 1). This indicates that all samples have a similar chemical potential to within 1 meV. In addition, we can conclude that the resistivity is sensitive to the bulk Fermi surface. For $\mathbf{B}\|\mathbf{a}$ the main frequencies are $F_\alpha = 26$ T, $F_\gamma = 34$ T, $F_{\gamma'} = 105$ T    $F_\delta = 147$ T    and    $F_{\gamma''} = 320$ T clearly indicating anisotropic 3D Fermi surface pockets. For most of the detected oscillation frequencies, we derive their cyclotron effective masses ($m^*$) of the carriers by fitting the temperature dependence of the oscillation amplitude (inset of Fig. 2b) with the Lifshitz–Kosevich formula (see ref. 39 and Supplementary Figs 5 and 6 as well as Supplementary Note 1). The values of the effective masses are $m_\alpha^* = (0.021 \pm 0.003)m_0$, $m_\beta^* = (0.05 \pm 0.01)m_0$ and $m_\gamma^* = (0.11 \pm 0.01)m_0$ for $\mathbf{B}\|\mathbf{c}$, whereas they are a factor of 4–10 greater for $\mathbf{B}\|\mathbf{a}$ with $m_\gamma^* = (0.13 \pm 0.03)m_0$, $m_{\gamma'}^* = (0.35 \pm 0.03)m_0$ and $m_\delta^* = (0.4 \pm 0.1)m_0$, where $m_0$ is the mass of a free electron (see Supplementary Table 1). These values are small and comparable to the effective masses in other slightly doped Dirac materials such as $Cd_3As_2$ or graphene[40,41]. These low masses, together with long scattering times are the reason for the high mobility and the huge transverse magnetoresistances seen in semimetals.

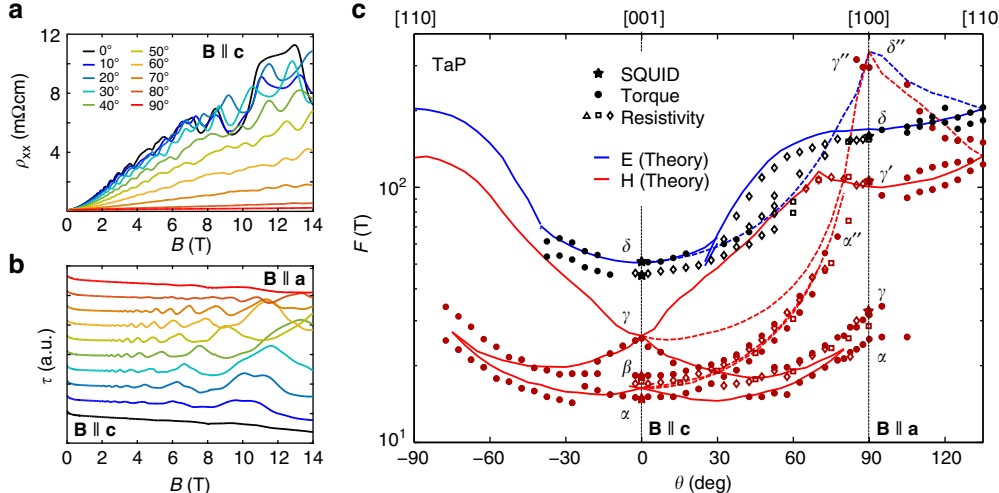

**Figure 1 | Quantum oscillations and angular dependence of oscillation frequencies in TaP. (a)** SdH oscillations in resistivity for different angles in steps of 10°. **(b)** dHvA oscillations from magnetic torque measurements for the same angles. Curves are shifted for clarity. **(c)** Full angular dependence of the measured and theoretical quantum oscillation frequencies. Open and closed symbols refer to SdH and dHvA data of five different samples from two different batches. Lines show the extremal orbits calculated from the banana-shaped 3D Fermi surface topology (solid lines for the pockets lying in the tilting plane of the magnetic field, dashed lines for the pockets lying perpendicular to it).

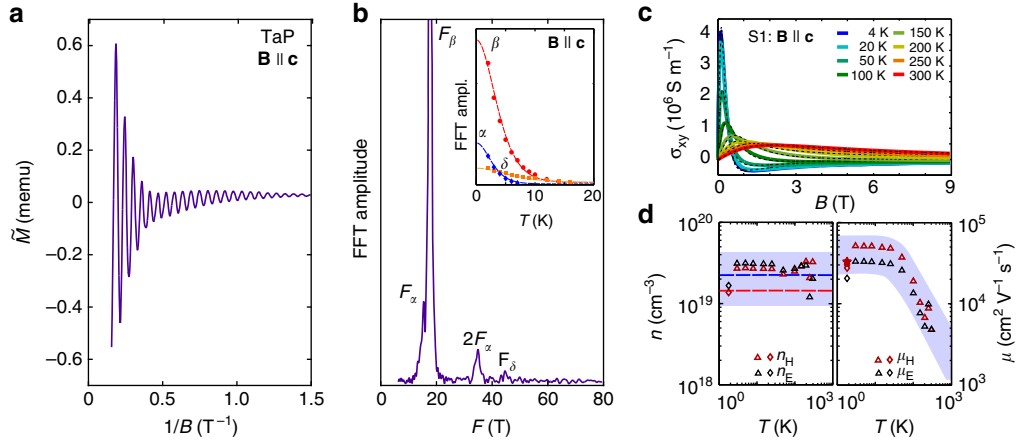

**Figure 2 | Charge carrier properties from quantum oscillation and Hall data. (a)** dHvA oscillations as a function of the inverse field at $T = 1.85$ K. **(b)** Fourier transform of **a** showing the characteristic quantum-oscillation frequencies. The inset shows the temperature dependence of the quantum oscillation amplitude and Lifshitz–Kosevich temperature reduction term fits. **(c)** Hall conductivity of sample S1 for different temperatures and two-band model fits (dashed lines). **(d)** Hole (H) and electron (E) carrier concentrations and mobilities as obtained by fitting the Hall conductivities of samples S1 (triangles) and S3 (diamonds), respectively. The grey-shaded areas give the confidence intervals of the densities and mobilities. The blue and red dashed lines mark the theoretical electron and hole densities based on the fitted Fermi-surface topology. The star marks the hole mobility determined from the Dingle analysis.

**Fermi-surface topology.** To reconstruct the shape of the Fermi surface, the full angular dependence of the quantum-oscillation frequencies was compared with band-structure calculations (Fig. 1c). The exact position of $E_F$ was determined by matching the calculated frequencies and their angular dependence to the experimental ones (see Supplementary Figs 7–10 and Supplementary Note 2). The best fit is obtained when $E_F$ lies 5 meV above the ideal electron–hole compensation point, in agreement with the resulting carrier concentrations from Hall measurements (Fig. 2c,d). At this $E_F$, calculations reveal two banana-shaped Fermi surface pockets, a hole pocket (H) and a slightly larger electron pocket (E). These two pockets reproduce the angular dependence of the measured dHvA frequencies with great accuracy (see lines in Fig. 1c and the Fermi surface in

Fig. 3). E and H are almost semicircular and are distributed along rings[5,6] on the $k_x = 0$ and $k_y = 0$ mirror-planes in the BZ. The rather isotropic frequencies $F_\alpha$ and $F_\gamma$ result from a neck and extra humps (head with horns) at the end of the hole pocket (see Fig. 3b). The splitting of all frequencies with field angles departing from **B∥c** in the (100)-plane is explained by the existence of four banana-shaped pockets in the BZ, two for each mirror plane for both E and H-pockets. The splitting of the frequency $F_\delta$ seen in the experiment is not reproduced by the calculation. One possibility for this discrepancy is that a waist may appear in the E-pocket at $k_z = 0$.

In addition, in the dHvA experiment the mobility of the hole orbits ($F_\alpha$ and $F_\beta$) was extracted via the width of the Fourier-transform peaks, which is given by the exponential

decrease of the oscillations with $1/B$ (the so-called Dingle term; see Supplementary Note 3). The deduced mobility is $\mu_h = 3.2 \times 10^4$ ($\pm 20\%$) $cm^2 V^{-1} s^{-1}$ (star in Fig. 2d).

**Charge-carrier density and mobility.** We extract information on the carrier density and mobility from magnetic field- and temperature-dependence of the Hall effect in sample S1 (full temperature range, Fig. 2c) and only at low temperature in sample S3 from a different batch. We employ a two-carrier model (see ref. 42) to fit the Hall conductivity ($\sigma_{xy}$) by making use of the longitudinal conductivity at zero field ($\sigma_{xx}$) as an additional condition (see Supplementary Note 4). As shown in the left panel of Fig. 2d, the carrier concentration for both electrons and holes at low temperature is around $n = (2 \pm 1) \times 10^{19}$ $cm^{-3}$ with an absolute error bar (grey-shaded area) approximately given by the difference between sample S1 and S3. Although TaP is an almost compensated metal the electron density is slightly larger than the hole density, which is in agreement with $E_F$ lying 5 meV above the charge-neutral point as determined from the Fermi surface topology. The theoretical values of the carrier densities, given as dashed lines in Fig. 2d, are in good agreement with the experimental data. Note that above 150 K the hole density becomes larger than the electron density. At low temperature, the mobility of both carriers is on the order of $\mu = (2-5) \times 10^4$ $cm^2 V^{-1} s^{-1}$, with the hole mobility higher than the electron one. These high mobilities indicate the very high quality of the single crystals with only few defects and impurities. The slight electron and hole mobility difference is also reflected in the sign change of the high-field Hall resistivity (see Supplementary Figs 11 and 12) and is similar to the Hall effect observed in TaAs[28,29] and NbP[30,36]. Furthermore, the Hall mobilities agree well with the mobility determined by the Dingle analysis (the star in Fig. 2d).

## Discussion

The experimental Fermi surface topology and charge-carrier concentrations described above converge to the same statement that the Fermi energy of our samples lies 5 meV above the ideal charge-neutral point in the calculated band structure. This slight carrier doping is not expected for a completely stoichiometric sample. However, the appearance of some defects/vacancies in this type of material is possible[43] and can explain the small shift of $E_F$. The consistency between experiment and theory strongly suggests that we found the true Fermi surface of our TaP samples. We plot the corresponding 3D Fermi surfaces from the *ab initio* calculations in Fig. 3. As can be seen, E and H are the only two pockets at the Fermi energy.

We shall further investigate the Weyl points in the band structure. In the 3D BZ, there are 12 pairs of Weyl points with opposite chirality: four pairs lie in the $k_z = 0$ plane (labelled as W1) and eight pairs are located in planes close to $k_z = \pm \pi/c$ (where $c$ is the lattice constant) (labelled as W2). One can see that the W1 points are far below $E_F$ by 41 meV and are included in the E-pocket. The W2-type Weyl points are 13 meV above $E_F$ and are included in the H-pocket. Such pairs of W2 points merge slightly into the head position of the H-pocket, leading to a two-horn-like cross-section (see $F_\gamma$ in Fig. 3b). Energetically, they are separated by a 16 meV barrier along the line connecting the Weyl points of a pair. We plot the energy dispersion of Weyl bands over the $F_\gamma$ plane in Fig. 3d. There are no independent Fermi-surface pockets around the W2 Weyl points and therefore the chiral anomaly is not well-defined. The Weyl cone is strongly anisotropic in the lower cone region below the Weyl point.

Finally, we discuss the longitudinal magnetotransport properties of TaP. We measured the longitudinal magneto-resistance of three samples from the same crystal batch. The current (**I**) was applied along the crystallographic **c** (sample S2)

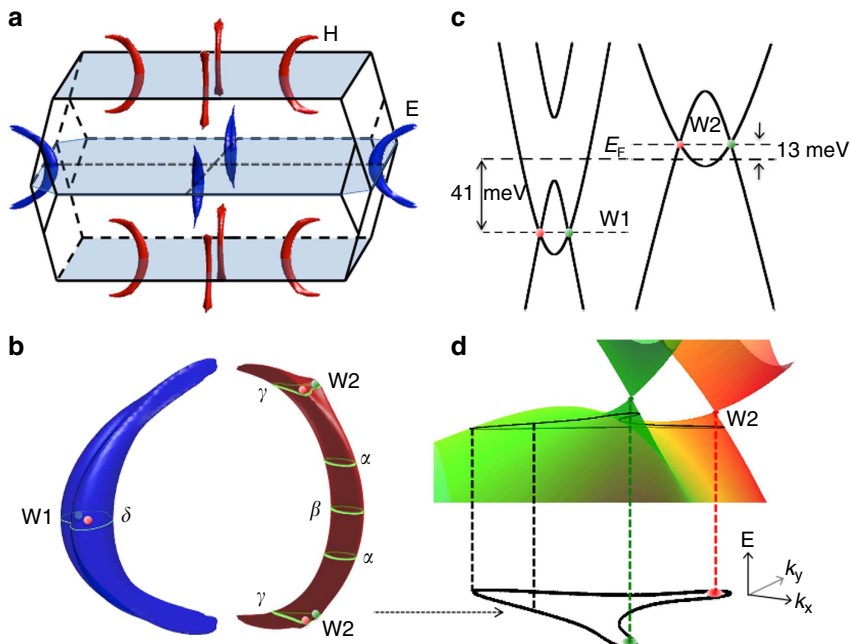

**Figure 3 | 3D Fermi surface pockets and Weyl points.** (**a**) Fermi pockets in the first BZ at the Fermi energy ($E_F$) detected in the experiment. The electron (E) and hole (H) pockets are represented by blue and red colours, respectively. (**b**) Enlargement of the banana-shaped E and H-pockets. The pink and green points indicate the Weyl points with opposite chirality. W1- and W2-type Weyl points can be found inside E and close to H-pockets, respectively. Green loops represent some extremal E and H cross-sections, corresponding to the oscillation frequencies measured, $F_{\alpha,\beta,\gamma,\delta}$ for **B**||**c**. (**c**) Energy dispersion along the connecting line between a pair of Weyl points with opposite chirality for W1 (left) and W2 (right). The deduced experimental $E_F$ (thick dashed horizontal line) is 13 meV below the W2 Weyl points and 41 meV above the W1 Weyl points. (**d**) Strongly anisotropic Weyl cones originating from a pair of W2-type Weyl points on the plane of $F_\gamma$. Green and red Weyl cones represent opposite chirality.

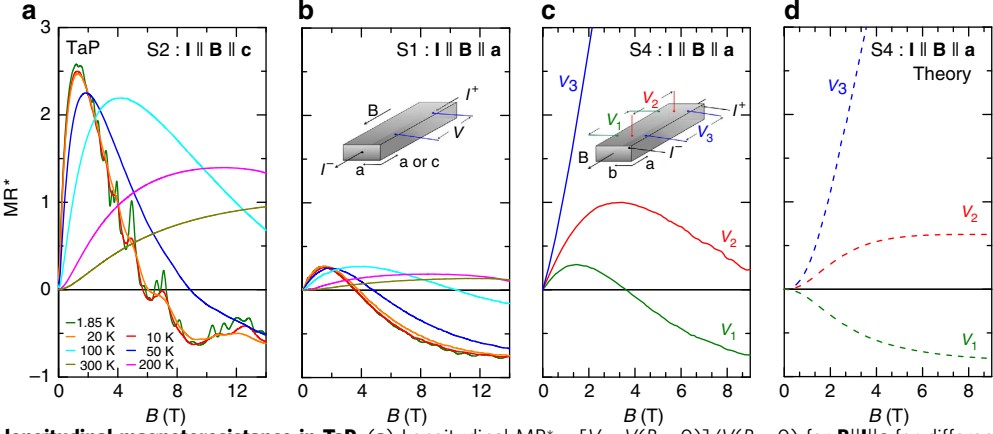

**Figure 4 | Negative longitudinal magnetoresistance in TaP.** (**a**) Longitudinal $MR^* = [V - V(B=0)]/V(B=0)$ for **B**||**I**||**c** for different temperatures. (**b**) Same for **B**||**I**||**a**. The temperatures are the same as in **a**. (**c**) Same for **B**||**I**||**a** and three pairs of contacts. The difference in the curves can be explained by an inhomogeneous current distribution induced by the magnetic field (see text). The contact geometry is shown in the inserts: (**b**) for S1 and S2, and (**c**) for S4. (**d**) Theoretical curves for S4 as in **c**.

and **a** axes (samples S1 and S4), respectively. Figure 4a,b represent the apparent longitudinal magnetoresistance ($MR^* = [V - V(B=0)]/V(B=0)$, where $V$ is the voltage drop at a fixed electric current) obtained for the two crystallographic directions. As long as the current flows homogeneously in the sample, $MR^*$ is equal to the usual $MR = [\rho - \rho(B=0)]/\rho(B=0)$. At 1.85 K, $MR^*$ first increases slightly and soon drops steeply from positive to negative values. The apparent negative MR is very robust against increasing temperature, and $MR^*$ is negative in a very narrow angular window of $\theta \leq 2°$ and is positive otherwise (see Supplementary Fig. 13 and Supplementary Note 5). This small $\theta$ window for the apparent negative MR is similar to that observed in Na₃Bi[27].

In light of the given Fermi-energy position in our crystals, it is not possible to link the negative MR to the chiral anomaly, simply because the former is not a well-defined concept when the Fermi surface connects both Weyl nodes. Although this does not rule out the presence of a non-trivial negative MR[44-46], we find that the observed negative $MR^*$, is strongly affected by the geometric configuration.

This becomes evident when we examine the apparent longitudinal MR for three different voltage contact configurations on sample S4 as illustrated in Fig. 4c. A clear voltage decrease in magnetic field is observed for pair V1, similar to the low temperature curves in Fig. 4a,b, whereas the two other pairs, denoted V2 and V3, show a higher $MR^*$. This points to an underlying inhomogeneous current distribution in the sample becoming important in high magnetic fields. As typical for high mobility semimetals, TaP has a large transverse MR arising from the orbital effect, whereas the longitudinal MR most likely stays of the same order of magnitude. For current contacts smaller than the cross-section of the sample, this leads to a field-induced stearing of the current to the direction of the magnetic field, which is along the line connecting the current contacts when current and magnetic field are parallel. This effect is known as current jetting[34,47-51]. As a consequence, a voltage pair close to this line (V3) detects a higher $MR^*$ than the intrinsic longitudinal MR whereas a voltage pair far away from it (V1) detects a smaller $MR^*$ than the intrinsic one. This effect is confirmed by calculations of the voltage distribution for sample S4, taking into account the current jetting by following ref. 51 (see Supplementary Figs 14 and 15 and Supplementary Note 6). Using the experimental transverse MR and assuming a field-independent intrinsic longitudinal MR, the model qualitatively reproduces the three $MR^*$ curves without any free parameters (see Fig. 4d). Therefore, the magnetic field

dependence of the longitudinal voltage is largely induced by the strong transverse MR, if the current is not homogeneously injected into the whole cross-section of the sample.

In summary, we determined the Fermi-surface topology of the inversion-asymmetric WSM TaP. The Fermi surface consists of banana-shaped spin-polarized electron and hole pockets with very light carrier effective masses. Despite the absence of independent Fermi-surface pockets around the Weyl points, an apparent negative longitudinal MR is detected. We show that in such studies, special care is needed to avoid a decoupling of the voltage contacts from the current jet in longitudinal magnetic fields.

## Methods

**Single-crystal growth.** High-quality single crystals of TaP were grown via a chemical vapour transport reaction using iodine as a transport agent. Initially, polycrystalline powder of TaP was synthesized by a direct reaction of tantalum (Chempur 99.9%) and red phosphorus (Heraeus 99.999%) kept in an evacuated fused silica tube for 48 h at 800 °C. Starting from this microcrystalline powder, the single-crystals of TaP were synthesized by chemical vapour transport in a temperature gradient starting from 850 °C (source) to 950 °C (sink) and a transport agent with a concentration of 13.5 mg cm⁻³ iodine (Alfa Aesar 99,998%)[52].

**X-ray analysis and structure characterisation.** The crystal structure and orientation of TaP crystals were determined by X-ray diffraction at room temperature. For this, TaP single crystals were mounted on a four-circle Rigaku AFC7 X-ray diffractometer with a built-in Saturn 724 + CCD detector. A suitable sample edge was selected where the transmission of $Mo - K_\alpha$ ($\lambda = 0.71073$ Å) radiation seemed feasible. The intensities of the obtained reflections were corrected for absorption by using a multi-scan technique. The unit cell was assigned by using a 30 images standard indexing procedure. Here oscillatory images about the crystallographic axes allowed the assignment of the crystal orientation, confirmed the appropriate choice of the unit cell and showed the excellent crystal quality. Supplementary Figs 1 and 2 show X-ray diffraction patterns of the S1 and S2 TaP crystals, which were used in our transport measurements. Structure refinement was performed by full-matrix least-squares on $F$ within the program package WinCSD, Version 4.14 (ref. 53 and revealed the non-centrosymmetric crystal structure with space group I4₁md and lattice parameters $a = b = 3.30$ Å, $c = 11.33$ Å at room temperature.

To confirm the quality of our TaP single crystals additional Laue images from an unperturbed as grown (001)-facet of a TaP crystal were taken. The single crystal was oriented using a white beam backscattering Laue X-ray diffraction method. Supplementary Fig. 3 shows the corresponding Laue diffraction image indexed with the I4₁md-structure and room temperature lattice parameters. The Laue diffraction image shows sharp reflections, which confirm the excellent quality of the sample. The presence of domains or twinning can be ruled out by indexing all reflections of the image by a single pattern.

**Quantum oscillations.** The electronic structure of TaP has been characterized by means of quantum oscillations. Here, the SdH and dHvA effect were measured by electrical resistivity, magnetization and torque magnetometry experiments[39]. The frequency of these oscillations $F = A_{ext} \times \hbar/2\pi e$ is proportional to the extremal

Fermi surface cross-section $A_{ext}$ perpendicular to the respective magnetic field direction[39,54,55]. Quantum oscillation spectra were obtained from magnetization and torque data by discrete Fourier transformation of the background subtracted oscillatory part of the respective signal.

**Magnetization and dHvA measurements.** The magnetism and magnetic quantum oscillations of TaP along the main crystallographic axes were investigated in a Quantum Design Inc. SQUID-VSM in the temperature and magnetic field range of 2–50 K and ±7 T.

Angular dependencies were measured using the Quantum Design Inc. piezo-resistive torque magnetometer (Tq-Mag[56]) in a physical property measurement system (PPMS) with installed rotator option. Magnetization and torque experiments were performed on two large 4.4 and 21.7 mg TaP single crystals. The samples were mounted on the sample holder and torque lever by GE varnish or Apiezon N grease and aligned along their visible crystal facets, which were confirmed by X-ray diffraction. The crystal alignment was verified by photometric methods and showed typical misalignments <2°. Magnetic torque measurements were performed up to 14 T in the same temperature range. Here the sample is mounted on a flexible lever which bends when torque is applied. The sample magnetization M induces a magnetic torque $\tau = \mathbf{M} \times \mathbf{B}$, which bends the lever and can be sensed by piezo-resistive elements which are micro-fabricated onto the torque lever. These elements change their resistance under strain and are thus capable of sensing the bending of the torque lever. The unstrained resistance of these piezo resistors is typically 500 Ω and can change by up to a few per cent when large magnetic torques are applied. Probing quantum oscillations, however, requires a higher torque resolution. Thus the standard torque option was altered and extended by an external balancing circuit and SR830 lock-in amplifiers as read-out electronic. This way a resolution of one in $10^7$ was achieved. The magnetic torque was determined for magnetic fields in the (100), (001) and (110)-plane in angular steps of 2.5° and 5°. Measurements were taken during magnetic field down sweeps from 14 to 0 T. The magnetic field sweep rate was adjusted such that the sweep rate in $1/B$ was constant. The resultant magnetic torque signals are a superposition of the sample diamagnetism, dHvA oscillation and uncompensated magnetoresistance of the piezo resistors. Because of the vector product of the magnetization and magnetic field, this method can only be applied to samples with strong Fermi surface anisotropy and is insensitive when the magnetic field and magnetization are aligned parallel, for example, along the crystallographic **c** direction in TaP.

The quantum oscillation frequencies are extracted from the obtained resistivities and magnetizations by subtracting all background contributions to those signals and performing a Fourier transformation of the residual signal over the inverse magnetic field. The resulting spectra show the dHvA frequencies and their amplitude.

**Magnetoresistance and SdH measurements.** Resistivity studies were performed in a PPMS using the DC mode of the AC-Transport option. Samples with two different crystalline orientations, that is, bars with their long direction parallel to the crystallographic **a** and **c** axes, were cut from large TaP single crystals using a wire saw. The orientation of these crystals was verified by X-ray diffraction. The samples were named S1 (**I**| **a**), S2 (**I**| **c**), S3 (**I**||**a**) and S4 (**I**| **a**). The physical dimensions of S1, S2, S3 and S4 are (width × thickness × length) 0.42 × 0.16 × 1.1 mm³, 0.48 × 0.27 × 0.8 mm³, 0.5 × 0.2 × 3.0 mm³ and 0.79 × 0.57 × 3.2 mm³, respectively. Contacts to the crystals were made by spot welding 25 µm platinum wire (S1 and S2) or gluing 25 µm gold wire to the sample using silver loaded epoxy (Dupont 6,838). The resistance and Hall effect were measured in six-point geometry using a current of about 3 mA at temperatures of 1.85–300 K and magnetic fields up to 14 T. Crystals were mounted on a PPMS rotator option. Special attention was paid to the mounting of the samples on the rotator puck to ensure a good parallel alignment of the current and magnetic direction. The Hall contributions to the resistance and vice versa were accounted for by calculating the mean and differential resistance of positive and negative magnetic fields. Almost symmetrical resistivities were obtained for positive and negative magnetic fields when current and magnetic field were parallel showing the excellent crystal and contact alignment of our samples. Otherwise, the negative MR* was overwhelmed by the transverse resistivity. To increase the sensitivity of the angular dependent SdH measurements at low magnetic fields, external Stanford Research SR830 lock-in amplifiers were used. Typical excitation currents of a 2–5 mA were applied at frequencies of ∼20 Hz.

**Band structure calculations.** The *ab initio* calculations were performed using density-functional theory with the Vienna *ab initio* simulation package[57]. Projector-augmented-wave potential represented core electrons. The modified Becke–Johnson exchange potential[58,59] was employed for accurate band structure calculations. Fermi surfaces were interpolated using maximally localized Wannier functions[60] in dense **k**-grids (equivalent to 300 × 300 × 300 in the whole BZ). Then angle-dependent extremal cross-sections of Fermi surfaces are calculated to compare with the oscillation frequencies according to the Onsager relation.

**Data availability.** The data that support the findings of this study are available from the corresponding authors B.Y. and E.H. upon request.

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

## Acknowledgements

We are grateful for K. Behnia, Y.-L. Chen, L.-K. Lim, Z.-K. Liu, E. G. Mele, J. Moore, S.-Q. Shen and D. Varjas for helpful discussions. This work was financially supported by the Deutsche Forschungsgemein- schaft DFG (Project No. EB 518/1-1 of DFG-SPP 1666 Topological Insulators, and SFB 1143) and by the ERC (Advanced Grant No. 291472 Idea Heusler). R.D.d.R. acknowledges financial support from the Brazilian agency CNPq.

## Author contributions

B.Y. conceived the project. E.H., B.Y., M.Ni. and C.F. supervised the project. F.A., M.Na. and M.B. carried out magnetization and magnetic torque measurements. C.S., R.D.d.R., N.K., M.O.A., F.A. and M.Na. performed magnetoresistance experiments. S.C.W., Y.S. and B.Y. calculated the *ab initio* band structure. F.A. carried out the quantum oscillation analysis and topology fitting. A.G.G. simulated the current density and potential distribution. M.S. grew the single crystals. D.S. and H.B. measured Laue and X-ray diffraction. F.A., C.S., A.G.G., J.H.B., M.Ni., E.H. and B.Y. wrote the manuscript. All authors contributed to the scientific planning and discussions.

## Additional information

**Competing financial interests:** The authors declare no competing financial interests.

