## [Peer Review File · Nature Communications]

Reviewer #1 (Remarks to the Author): The article by Arnold, Shekhar and collaborators now focusses almost exclusively on the quantum oscillation results, which demonstrate that the chemical potential is sufficiently far from the Weyl nodes, that no Fermi surface can be considered chiral. This new perspective brings significant clarity to the manuscript. The somewhat unfortunate by product is that the interesting claim of negative magnetoresistance as a consequence of the chiral anomaly is no longer valid. Instead the authors show that the result is dominated by an experimental artifact known as current jetting. Scientifically, I quite like the revised version. The quantum oscillation study is excellent and provides a very comprehensive picture of the electronic structure. The authors even at one point acknowledge when small differences between theory and experiment exist. The cynic would argue that the lack of new physics precludes this from publication in a high profile journal. However, I would argue that we should welcome such a detailed study, and even if the results are perhaps less exciting from the viewpoint of new physics, this type of careful study should be encouraged. Hence, I recommend publication of the manuscript in Nature Communication.

Reviewer #2 (Remarks to the Author): The authors have made many revisions following the referees' reports and the revised manuscript now focuses on the quantum oscillations and related Fermi surface topology which is also very interesting and important. The title "Negative magnetoresistance without well-defined chirality in the Weyl semimetal TaP" which still emphasizes on the chirality may not be suitable. Moreover, I do not think the statement "Our results provide a clear framework how to detect the chiral magnetic effect" (in the abstract) is supported by the paper.

Reviewer #1 (Remarks to the Author): no comment.

Reviewer #2 (Remarks to the Author): We still believe this title to be suitable because it contains both information on the Fermi surface topology and the magnetoresistance, the two main results of our study. This sentence was removed when the abstract was shortened.